# Structural and Optical Properties of Self-Catalyzed Axially Heterostructured GaPN/GaP Nanowires Embedded into a Flexible Silicone Membrane

**DOI:** 10.3390/nano10112110

**Published:** 2020-10-23

**Authors:** Olga Yu. Koval, Vladimir V. Fedorov, Alexey D. Bolshakov, Sergey V. Fedina, Fedor M. Kochetkov, Vladimir Neplokh, Georgiy A. Sapunov, Liliia N. Dvoretckaia, Demid A. Kirilenko, Igor V. Shtrom, Regina M. Islamova, George E. Cirlin, Maria Tchernycheva, Alexey Yu. Serov, Ivan S. Mukhin

**Affiliations:** 1Nanotechnology Research and Education Centre of the Russian Academy of Sciences, Alferov University, Khlopina 8/3, 194021 Saint Petersburg, Russia; burunduk.uk@gmail.com (V.V.F.); acr1235@mail.ru (A.D.B.); fedina.serg@yandex.ru (S.V.F.); azemerat@rambler.ru (F.M.K.); vneplox@gmail.com (V.N.); sapunovgeorgiy@gmail.com (G.A.S.); liliyabutler@gmail.com (L.N.D.); george.cirlin@mail.ru (G.E.C.); 2Department of Chemistry, Peter the Great Saint Petersburg Polytechnic University, 195251 St. Petersburg, Russia; zumsisai@gmail.com; 3School of photonics, ITMO University, Kronverksky Prospekt 49, 197101 Saint Petersburg, Russia; imukhin@yandex.ru; 4Ioffe Institute, Politekhnicheskaya 29, 194021 St. Petersburg, Russia; 5Institute for Analytical Instrumentation of the Russian Academy of Sciences, Rizhsky pr. 26, 190103 St. Petersburg, Russia; igorstrohm@mail.ru; 6The Faculty of Physics and the Institute of Chemistry, Saint Petersburg State University, Universitetskaya Emb. 7/9, 199034 St. Petersburg, Russia; r.islamova@spbu.ru (R.M.I.); a.serov@spbu.ru (A.Y.S.); 7Centre of Nanosciences and Nanotechnologies, UMR 9001 CNRS, University Paris Sud, University Paris-Saclay, 10 Boulevard Thomas Gobert, 91120 Palaiseau CEDEX, France; maria.tchernycheva@c2n.upsaclay.fr

**Keywords:** flexible optoelectronics, self-catalyzed, dilute nitrides, GaPN, GaP, nanowire, NW membrane: III-V on Si, PDMS, diluted nitride, axially heterostructure

## Abstract

Controlled growth of heterostructured nanowires and mechanisms of their formation have been actively studied during the last decades due to perspectives of their implementation. Here, we report on the self-catalyzed growth of axially heterostructured GaPN/GaP nanowires on Si(111) by plasma-assisted molecular beam epitaxy. Nanowire composition and structural properties were examined by means of Raman microspectroscopy and transmission electron microscopy. To study the optical properties of the synthesized nanoheterostructures, the nanowire array was embedded into the silicone rubber membrane and further released from the growth substrate. The reported approach allows us to study the nanowire optical properties avoiding the response from the parasitically grown island layer. Photoluminescence and Raman studies reveal different nitrogen content in nanowires and parasitic island layer. The effect is discussed in terms of the difference in vapor solid and vapor liquid solid growth mechanisms. Photoluminescence studies at low temperature (5K) demonstrate the transition to the quasi-direct gap in the nanowires typical for diluted nitrides with low N-content. The bright room temperature photoluminescent response demonstrates the potential application of nanowire/polymer matrix in flexible optoelectronic devices.

## 1. Introduction

Heterostructured III-V nanowires (NWs) are of great interest for both practical application and fundamental research point of view providing new information about the crystal formation at the nanoscale. Unique NW geometry with a high surface to volume ratio, on one hand, promotes effective strain relaxation in lattice mismatched material systems resulting in suppression of the structural defects formation, which can act as non-radiative charge carrier recombination centers limiting efficiency of optoelectronic devices [1], and, on the other hand, allows for stabilization of metastable alloy compositions [2] and polymorph phases [3]. Moreover, geometry and high refractive index of III–V semiconductor NWs can lead to appearance of waveguide and cavity effects allowing more effective light scattering and collection compared to the thin films [4,5,6]. Thus III–V semiconductor NWs were considered as a promising material for light-emitting and light-detecting applications [7,8,9,10].

In addition, NW aspect ratio allows them to withstand high mechanical stresses [11], which makes them a suitable material for nanoscale and flexible optoelectronic devices—one of the main goals of the modern semiconductor technology [12,13,14]. One of the recent [15,16] approaches to fabricate a flexible optoelectronic device based on free-standing NWs is to embed a NW array into a transparent polymer matrix and remove composite NW/polymer membrane from the wafer [17,18,19]. Nowadays, materials based on III–V NWs encapsulated into flexible polymer membranes are of great interest in a wide range of applications. Several important implementations including: solar cells [9,10,20,21], infrared laser radiation visualizer [22], light-emitting devices (LEDs) [7,8,23,24,25], terahertz modulators [26,27], thermoelectric device [15], and other flexible devices [28] were reported recently.

Flexible LEDs on the base of organics (OLEDs) are the most commercially developed devices. Their properties, such as efficient electroluminescence (EL) and relatively easy and inexpensive fabrication allowed the OLED-based industry to cover a significant market. For instance, touchscreens of smartphones are mostly produced with the OLED displays [29]. However, organics are far behind inorganic semiconductors in terms of stability and external quantum efficiency of EL in visible spectral range [30,31]. Inorganic LEDs on the base of binary and ternary compounds of phosphides, arsenides, and nitrides, etc., are considered to be the materials for high brightness–long lifetime LEDs [32,33,34], which has quickly been developing these last years.

However, it should be noted that it is a challenge to adapt A3B5 thin film technology for high resolution RGB displays, because it requires both combination of different crystalline materials and advanced post-growth processing, including epitaxial layers lift-off from the rigid growth substrates [32].

One of the main advantages of NW-based devices is the possibility to combine materials with different chemical compositions, e.g., phosphides, nitrides, and arsenides. Polymer membranes with encapsulated NWs can be stacked onto each other in order to produce RGB displays with multiple lines of EL, where different colors correspond to different material systems. Thus, NW-based membrane devices can be considered as inorganic analogue to OLEDs.

Being an indirect bandgap III–V semiconductor, GaP has one of the broadest transparency range and exhibits one of the highest values of nonlinear refractive index among other III–V materials, which causes an interest of its application as a material for nonlinear nanophotonic devices [22,35,36]. Functionality of GaP based nanomaterials can be broadened by using ternary GaPN dilute nitride alloys demonstrating quasi-direct bandgap behavior and giving the possibility of bandgap engineering in the range of (1.2–2.15) eV [37,38]. The growth of self-catalyzed core-shell GaP/GaPN NWs were demonstrated and their various properties were extensively studied [39,40,41,42]. However, formation of axially heterostructured diluted nitride NWs seems illusive due to the reasons below. Sukrittanon and Tu [39] study the effect of nitrogen incorporation on the crystal quality and formation of diluted nitride layers and NWs. It was shown that the presence of nitrogen flux from the very beginning of the NW formation fails the Vapor–Liquid–Solid (VLS) growth mechanism and no NWs are nucleated, thus, the authors demonstrate only formation of core-shell dilute nitride NWs [39]. However, as will be shown further in the present report, VLS formation of axially heterostructured diluted nitride NWs can be achieved by using GaP NW stem.

Commonly metalorganic vapor-phase epitaxy (MOVPE) and gas-source molecular beam epitaxy (GS-MBE) [43,44] techniques are used for the GaPN NWs growth. MOVPE is scalable and allows effective suppression of a parasitic two-dimensional layer formation. However, there are several disadvantages of the MOVPE technique, namely, a problem of oxygen and carbon contamination, parasitic radial growth resulting in structural heterogeneity, which can negatively affect the optical properties of the NWs due to a large amount of non-radiative recombination centers [40]. However, there are no reports concerning growth of dilute nitride NW via plasma-assisted molecular beam epitaxy (PA-MBE) approach, which can be beneficial as no additional hydrogen is presented on a growing surface and low growth temperatures are accessible [45,46].

Despite a large number of works dedicated to growth and study of diluted nitride NWs, several fundamental questions are still unraveled. With the main being, whether it is possible to obtain VLS growth of the GaPN. Here we focus on the PA-MBE growth of axially heterostructured diluted nitride GaP/GaPN NWs and investigation of their structural and optical properties. To systematically study the photoluminescent properties of the NWs and demonstrate their perspective for flexible applications, we encapsulate the NWs in the silicone membrane.

## 2. Materials and Methods

The experimental section is organized in the order of sample processing and study. First, in Section 2.1, we describe the epitaxial growth of NW arrays. Afterward, the technological steps of NW encapsulation into the silicon rubber membrane, and further release from the growth substrate, are presented (Section 2.2). The characterization methods are listed in Section 2.3.

### 2.1. MBE Growth of NWs

GaPN/GaP NW heterostructure arrays were grown on vicinal (4°) silicon (111) substrates within a self-catalyzed VLS approach by plasma-assisted molecular beam epitaxy (PA-MBE) using Veeco GEN-III MBE machine (Plainview, NY, USA). The setup was equipped with the Riber radio frequency--plasma (13.56 MHz) assisted source of activated nitrogen and phosphorus valved cracker. Before the growth, the Si substrates were cleaned by the modified Shiraki technique [47] finalized by the boiling in 35% nitric acid water solution to form a surface oxide layer in a controllable manner [48]. After loading and degassing, the substrates were annealed at 870 °C, which is 10 °C lower than the temperature of thermally deoxidizing of Si. Similar to the GaAs NW formation, silicon surface oxide condition was found to be a crucial parameter on the GaP vertical NW yield [49]. The chosen procedure of silicon oxidation and surface oxide annealing made it possible to maximize the NW surface density.

As shown in [50], the growth parameters have an influence on the NW array morphology, such as surface density, NW length, and diameter. The growth temperature was controlled by both thermocouple and pyrometer and was fixed at 630 °C. Ga flux was kept constant for all samples and was equivalent to 0.27 mL/s growth rate of planar GaP/Si (111). Phosphorus flux was set 3 times higher than stoichiometric P/Ga flux ratio for GaP formation and was kept constant during the NWs growth. P/Ga flux ratio was chosen according to our previous studies and believed to be optimal to obtain stable self-catalyzed formation of dense GaP NW array with high NW aspect ratio (l/d ~ 50–100) [22,51]. According to our observations, higher growth temperature and the V/III ratio values lead to the increasing of the vertical NW yield [51]. Notably, V/III flux ratios higher than 4 lead to the catalytic droplet shrinkage and interruption of the VLS growth mechanism. [22]. Thus, chosen growth conditions allow to preserve VLS growth mechanisms during the growth of diluted nitride NW segments. Diluted nitride NW segments of the GaPN/GaP NWs were grown at low N2 flux and high-power density of the RF-plasma, keeping the N_2_ gas flow at 0.5 sccm and input RF-power at 450 W, correspondingly. Both gallium and phosphorus fluxes were not interrupted during the N plasma ignition for the growth of heterostructured NWs.

The growth of GaP NWs was ended by closing Ga shutter and cooling the sample at the rate of 30 °C/min, maintaining the phosphorus flux value reduced three times until 400 °C to avoid GaP thermal decomposition. The growth of heterostructured GaP/GaPN NWs was ended by simultaneous interruption of all molecular fluxes to preserve the catalytic droplets.

### 2.2. Silicone Rubber Membrane Fabrication

In this work, polydimethylsiloxane (PDMS) “Sylgard 184” was used as a transparent silicon rubber source. NWs array was encapsulated into a PDMS matrix using the novel G-coating method (see Figure 1). This approach is based on the use of a swinging-bucket centrifuge centripetal force to achieve uniform coverage of the well-developed structure surface by PDMS rubber. To process samples, we used a high-speed Eppendorf Centrifuge 5804 R (Edison, NJ, USA).

During the encapsulation process PDMS base was mixed with a curing agent (10:1 mass ratio) and dropped onto the samples of 2 cm^2^ size and G-coated at 5000 rpm for 40 min. After PDMS deposition, the samples were cross-linked in a muffle oven at 80 °C for 8 h. The PDMS/NW membrane was released from the silicon substrate with the use of a thick PDMS support layer (50–150 µm), denoted as cap-film. This layer was necessary for the mechanical stability of the thin NW-encapsulated membrane (see Figure 1b). Then, as shown in Figure 1c, the polymer membrane was released from the Si substrate. The NW encapsulation and exfoliation technique employing cap-film for mechanical support was described by Neplokh et.al. in [52,53].

### 2.3. Sample Characterization

The morphology of NWs was studied using scanning electron microscopy (SEM) (Zeiss SUPRA 25) (D-73446, Oberkochen, Germany). Crystal structure was studied with transmission electron microscopy (TEM) using JEOL JEM–2100F field emission gun TEM (Tokyo, Japan) operating at 200 kV (point-to-point resolution of 0.19 nm in TEM mode). For TEM studies NWs were “dry-transferred” to a carbon coated TEM grid by sliding the grown sample on a TEM grid face to face. The compositional analysis was performed with scanning transmission electron microscopy (STEM) (Tokyo, Japan) in electron energy loss spectroscopy (EELS) and energy dispersive spectroscopy (EDS) modes [54].

The measurements from as-grown NW arrays and individual NWs were performed at room temperature (RT, 300K) using a Horiba LabRAM HR800 spectrometer (Kyoto, Japan) and an optical microscope for excitation and collection in backscattering geometry. The measurements were carried out with a λ = 532 nm diode-pumped solid-state (DPSS) YAG:Nd (neodymium-doped yttrium aluminum garnet) continuous wave laser providing near bandgap excitation of GaP and GaPN solid alloys. The local photoluminescence (µ-PL) measurements from individual NWs were obtained using an ×100 collecting objective. The excitation beam was focused on the sample to a spot of ≈1 µm diameter.

The optical measurements from arrays of NWs at the temperature of 5 K were performed with the Melles Griot laser Model: 85-GLS-301 (Rochester, NY, USA) an excitation wavelength of λ = 507 nm using a closed cycle He cryostat. The spectrometer was based on an MDR-204-2 monochromator LOMO FOTONIKA (Saint-Petersburg, Russia) and a Hamamatsu R298 photomultiplier tube (Hamamatsu, Japan). These measurements were performed for as-grown NWs and NWs encapsulated into a silicone rubber membrane in order to distinguish NW PL response apart from parasitically grown layer.

## 3. Results and Discussion

To study the effect of the active nitrogen flux on the NW growth mechanism, we synthesized the series of samples: (i) GaP NWs stem grown for 1 h; (ii) heterostructured NWs consisting of two segments with an equal growth time (1 h) with a GaPN segment grown on a GaP stem (grown similarly to the sample 1); and (iii) a reference GaP NW sample grown similar to the sample (ii) but without the introduction of activated nitrogen molecular flux. Expected sample structures are presented in Figure 2a.

First, let us discuss the effect of the nitrogen flux on the self-catalyzed VLS NW growth mechanism. Side-view cleaved edge SEM images of the synthesized samples are presented in Figure 2b–d. Both vertically-oriented NWs and unwanted 3D parasitic islands are formed in all samples and are depicted in Figure 2 as green rods and magenta polygons, respectively. Both NWs and parasitic islands have similarly oriented in-plane hexagonal cross-sections. It should be noted that despite the introduction of the active nitrogen flux, the height of heterostructured GaPN/GaP NWs increases proportionally to the growth time. As can be judged from the statistical analysis of the NW arrays morphology, both mean heights and diameters of the pure GaP and heterostructured GaPN/GaP NWs are identical. The average length of the studied GaP NWs stem-sample is 3.3 µm, of the reference GaP sample—about 6.6 µm and heterostructured GaPN/GaP NWs is 6.2 µm.

The average diameters of NWs are about 95, 125, and 120 nm for the stem-sample, reference GaP NW, and GaPN/GaP NWs, respectively. Surface density: stem-sample—0.58 μm^−2^, reference GaP NWs—1.3 μm^−2^ and GaPN/GaP NWs—0.97 μm^−2^. Since the surface density of the NWs stem-sample is almost two times lower compared to the surface density of the other synthesized NW arrays, it can be concluded that the nucleation of NWs occurs not only at the very beginning of the synthesis, but also during the growth process. One can suggest that the introduction of nitrogen flux can inhibit NW nucleation as it can fail the self-organized formation of new catalytic droplets [39], as NW surface density is lower in heterostructured NW sample.

Notably, the remaining Ga catalyst droplets on the GaP NWs tips (marked with orange color in Figure 2) is the evidence of the self-catalyzed growth mechanism. Earlier, self-catalyzed growth of axial NW heterostructures based on the group V interchange was observed in GaAs/GaP [51] and InSb/InAs [55] material systems. The difference in catalytic droplet size is related to the droplet consumption in the pure GaP samples during the cooling under the phosphorus flux in comparison to the GaPN/GaP sample, which was cooled in ultrahigh vacuum conditions. Thus, we conclude that the active nitrogen flux values used in this work did not have a significant effect on the self-catalyzed VLS growth mechanism. We assume that earlier attempts on a diluted nitride NWs synthesis via a self-catalyzed approach have failed due to the fact that the presence of an active N-flux suppressed the formation of a proper array of catalytic Ga droplets. Thus, if the GaP NW stem has been formed, NW growth can be continued under N-flux. In fact, numerous works devoted to the formation of the self-catalyzed GaN NWs indicating that nitrogen can be dissolved in liquid Ga droplet were reported previously [55,56,57,58].

The incorporation of N species during the diluted nitride NW growth is rather questionable. One can suggest that N does not incorporate into NW lattice at all or axial heterostructures are formed. As will be shown, a further incorporated nitrogen fraction can be estimated by the optical spectroscopy. However, we should note that radial growth occurs in both pure GaP reference and GaPN/GaP heterostructured samples together with axial growth. We estimate the thickness of the shell formed due to the radial growth to be equal to 15 nm. The proposed structure of NW heterostructures schematically shown can be seen in Figure 2a.

### 3.1. Crystal Structure Study

Reflection high energy electron diffraction (RHEED) pattern acquired in-situ, shown in Figure 3c, demonstrate that NWs have a zinc-blende structure and grow coherently to a Si(111) wafer lattice, with the exception of 180° rotation twinning along the growth direction. RHEED Bragg peaks have a horizontal dash shape due to the refraction effect in transmission diffraction [59,60]. Thus, we can conclude that the NW arrays are regularly shaped. Dark-field TEM (DF-TEM) imaging with a diffraction contrast was chosen to provide information about the NW structural and phase homogeneity and growth defects. NWs were oriented on the [110] _ZB_ zone axis along the e-beam, which allows stacking faults and lattice twining to be distinguished. Typical DF-TEM images of the individual NWs obtained with the wurtzite reflex are presented in Figure 3a,b.

TEM analysis exhibits that both GaPN/GaP and reference GaP NWs possess high density of planar defects, mainly rotational twin boundaries, which appear as thin bright lines crossing the NW. These lines are randomly spread across the NW length (see Figure 3). As mentioned above, one of the main evidence of the self-catalyzed growth of GaPN/GaP NWs is the preservation of a Ga droplet at the NW tip (marked in orange color in Figure 2b–d and Figure 3a,b), which is clearly visible for the GaP/GaPN sample cooled in the vacuum condition. In contrast, sample cooled under the phosphorus flux demonstrates the presence of thin NW segments of hexagonal wurtzite (WZ) phase [61] formed at the NW tip under the shrinking Ga-droplet. Note, WZ phase also appears at NW base for all samples. These insertions are depicted with white arrows (see Figure 3a,b). We assume that the formation of the WZ phase in the top part of the NW is associated with the consumption of a catalytic Ga droplet during the sample cooling under the excess phosphorus flux and subsequent decrease in the contact angle favoring the nucleation on the triple VLS line [62]. Note, as shown in [1], and will be discussed further, the existence of inserts of the metastable WZ phase should not significantly affect the optical properties of GaP/GaPN NWs.

Comparison of high-angle annular dark-field imaging (HAADF) (Figure 3e) and DF-TEM images (Figure 3a,b) of the heterostructured GaPN/GaP and pure GaP NWs shows no evidence of any visible contrast, which can indicate the possible formation of a GaPN/GaP heterointerface. Comparing the reference sample with heterostructured GaPN/GaP NWs and GaP stems, no significant difference in the crystal structure and crystal perfection can be distinguished. Apparently, a small change in the GaPN lattice constant with small concentration of nitrogen (<1%) is insufficient for its detection by electron microdiffraction or EELS. This may indicate that the incorporation of a small amount of the nitrogen atoms into the GaP matrix has practically no effect on the crystal structure and perfection of the synthesized NWs. In addition, we were not able to detect any prominent signal from nitrogen species by both EELS and EDS techniques in STEM analysis.

### 3.2. Optical Properties Study

Obviously, the analysis of the as-grown NWs arrays with integral evaluation techniques is not straightforward due to the presence of parasitic islands. Thus, we perform (i) optical studies of individual NWs mechanically transferred to the quartz glass substrate; and (ii) NW arrays embedded into transparent silicone rubber released from the growth substrate; and (iii) parasitically grown island layer on Si substrate.

### 3.3. NW Composition (μ-Spectroscopy PL and Raman Study)

The microspectroscopy studies of the PL response and Raman scattering (RS) of individual NWs were performed at 300K to investigate the spatial distribution of the NW chemical composition (see Figure 4). It is known that additional Raman active modes commonly appear due to alloying effect. Thus, diluted nitrides segments of the heterostructured NW can be distinguished by the appearance of the additional RS modes [40,63].

We find that NWs consist of two segments which differ by both the RS signal intensity and appearance of additional modes. Typical RS spectra obtained at the opposite ends of a single heterostructured NW, depicted as (1) and (2), are presented in Figure 4c. The spectrum obtained from the point (2), shown in Figure 4c, corresponds to the pure GaP Raman spectrum, and possesses three pronounced bands, typical for GaP in the NWs geometry [40], namely the transverse optical (TO-), longitudinal optical (LO-) phonon modes, corresponding to the zone-center (Г) optical phonons in ZB GaP, and surface optical (SO) phonon mode, that can be activated by the breakdown of translational symmetry at the NW surface. The observed modes are located at 365, 402, and 394 cm^−1^, correspondingly [41]. Thus, we associate the bottom NW segment with a GaP NW stem. On the contrary, as can be seen in spectrum (1) obtained in the NW upper segment, an additional vibrational mode appears. This mode (labeled as X) is located between TO and LO GaP-like Raman bands and can be attributed to the mentioned above disorder-activated optical phonons occurring due to N clustering and alloy inhomogeneity [64]. A slight shift of the Raman band position also proves the N atoms incorporation [37,65].

Spatial distribution maps of the integrated Raman scattering intensity of the TO and X-mode from individual NW are shown in Figure 4a,b, correspondingly. Heterogeneous distributions of TO- and X-Raman bands intensity along the NW are clearly visible. One can note that TO-mode intensity is increased in a NW upper segment. We suggest that the overall increase of TO-mode intensity together with an appearance of diluted nitride specific X-mode can be associated with an increase in absorption cross-section in direct-bandgap GaPN material. Thus, we suggest that the absence of the X-alloy Raman band at spectrum 2 is the evidence of the absence of the incorporated N atoms into the GaP stem of heterostructured NW.

The micro-photoluminescence (μ-PL) study shows the broad RT PL emission from 540 to 700 nm spectral range. Figure 4b demonstrates the μ-PL spectrum taken from the brightest spot from the individual heterostructured GaPN/GaP NWs. Since the PL maximum peak position is located at 610 nm, the average N content into NW can be estimated as high as 1% [66]. It is worth noting that the spatial map of the PL signal intensity demonstrated in the inset in Figure 4d) follows the same dependencies as RS (see Figure 4a). The brighter signal corresponds to the GaPN ternary alloy direct gap NW segment. The complete absence of the PL signal on the GaP NW segment proves the absence of embedding of nitrogen atoms, and, therefore, is another confirmation of the dominant role of axial growth mechanism in the formation of heterostructured NWs. In addition, one can note bright spots at X-mode and PL spatial maps at the NW top and bottom facets. We suggest that the appearance of the RS and PL hot spots is caused by the NW waveguiding effect and more efficient light scattering and coupling at the NW edges. Thus, one can conclude that heterostructured GaP/GaPN nanowires are axially heterostructured.

### 3.4. Photoluminescent Properties

#### Membrane Release Description

In order to determine the photoluminescence properties of the synthesized arrays of NWs and the parasitic layer separately, we studied the released membrane and as-grown NW array. After membrane exfoliation, SEM imaging was used to clarify how exactly NWs were encapsulated into polymer membrane (see Figure 5f) and then were exfoliated from Si substrate (see Figure 5d).

As can be seen in Figure 5d, after membrane releasing, two main areas on the substrate can be distinguished: (i) the area with completely removed NWs and (ii) the area with partially preserved NWs (labeled in green color in Figure 5b,d,f). Generally, only parasitic islands (labeled with magenta polygons in Figure 5d) remain on the substrate after membrane releasing. However, hexagonal NW stumps remain at the substrate (depicted as blue hexagons in Figure 5d). The surface density of stumps (0.8 cm^−2^) is approximately equal to the surface density of as-grown NWs (0.97 cm^−2^).

As shown in Figure 5f, the surface of the polymer membrane with encapsulated GaPN/GaP NWs is rough. The main contribution to the membrane surface roughness is made by the imprint of the parasitic islands, which is depicted with red polygons in Figure 5f. The pronounced hollows on the membrane surface are the traces of unreleased parasitic islands. Note, the NWs bases in the form of needles on the membrane surface are clearly observed (see Figure 5f). Thus, the parasitic 2D layer is practically not transferred into the polymer matrix. The proposed approach allows to separate NWs array and 2D parasitic layer for further optical measurements. The optical radiation collected from the NWs-encapsulated membranes can be attributed only to the PL signal from NW arrays.

Figure 5a demonstrates the PL response measured on the as-grown pure GaP and heterostructured GaPN/GaP NW arrays on Si(111). PL spectra obtained from pure GaP NWs (reference sample) are shown with blue line in Figure 5, and expectably demonstrate the negligible PL signal due to the indirect-gap transition in zinc-blende GaP [67,68]. We should note that we do not observe any PL emission related to the presence of WZ inclusions, which is consistent with earlier observations suggesting quasi-direct bandgap behavior of WZ GaP phase with dipole-forbidden direct transition [69,70]. In contrast, the dilute nitride samples demonstrate bright PL signal indicating the quasi-direct and direct bandgap transitions [67]. Despite high density of structural defects, we observe reliable PL signals at both 300K and 5K. It was shown earlier that rotational twinning defects can act as an effective non-radiative recombination center and lead to a quenching of PL emission intensity at low temperatures, while at 300K PL quenching occurs due recombination via point defects [1].

PL spectra, acquired at the Si substrate with as-grown NWs and the substrate after membrane exfoliation demonstrate the presence of two broad peaks with fine structure (see Figure 5a,c). A set of sharp peaks can be attributed to highly-localized N-levels (and their phonon replica) in parasitic islands, where NN_i_ is the position of highly-localized N levels in diluted nitrogen GaPN solid alloys (see Figure 5a,c). The nature of these peaks is associated with the recombination of carriers through localized states formed by the closest pairs of nitrogen atoms. The detailed description of these states was reported on by Thomas [71] and Lazarenko [72]. The presence of peaks on PL spectra is the evidence of a quasi-direct bandgap of GaPN solid alloys [67]. Relatively high intensities of NN_3_ and NN_4_ peaks indicate the low average content of N atoms diluted into GaPN parasitic islands and also show the quasi-direct bandgap of alloy. The concentration of nitrogen impurities can be estimated as 0.2–0.3% [67,73,74,75,76].

On the contrary, PL signal from strongly localized levels appears to be quenched in the PL spectra obtained from the exfoliated NW/PDMS membrane, while the broadband PL response appears to be slightly red shifted—see Figure 5e. A comparison of the absolute intensities of the obtained PL spectra suggests that the emission of parasitic island layer can prevail over the emission from the NWs in PL response from the as-grown epitaxial structure due to the smaller volume of the latter. The PL spectrum observed from GaPN/GaP NW-encapsulated PDMS membrane has a complex shape and the broad PL signal at the spectral range from 560 to 660 nm with (full width at half maximum) FWHM of 55 nm and maximum located at 586 nm. The broadband character of the PL response and red-shifted PL maximum position indicate that the concentration of diluted nitrogen atoms in the GaPN/GaP NWs is about 1% [74], which is higher than the estimated value for the parasitic islands.

Thus, we can conclude that nitrogen incorporation occurs more effectively during VLS NW growth in comparison with three dimensional Volmer–Weber growth of island layer via vapour-solid (VS) mechanism. According to the papers devoted to the formation of self-catalyzed GaN NW nitrogen can be dissolved in the catalytic droplet at NW growth temperature [56,57,58,77]. Thus, nitrogen incorporation depends on the thermodynamic of GaPN liquid solution and component solubility. In contrast, nitrogen incorporation during VS growth can be limited by the over stoichiometric (x3) phosphorus flux [78]. We assume that catalytic droplet adsorbs N atoms in the same way as other group-V elements due to the impingement molecular flux and due to surface diffusion along NW side-facet. As mentioned above, NW growth occurs not only in axial via VLS mechanism, but also in radial direction via VS or mixed VLS-VS (then catalytic droplet inflation leads to NW radius increase) mechanisms. However, observed radial growth rate during formation of the top NW segment is 200 times lower than axial growth rate (3 μm/h vs. 15 nm/h). Thus, assuming low efficiency of nitrogen incorporation during VS growth we can propose that an N-containing NW shell has a minor effect on the NW optical properties.

## 4. Conclusions

The growth technique allowing the formation of axially heterostructured GaPN/GaP NW on Si (111) via PA-MBE with N-content up to 1% is reported. Analysis of the morphology of the heterostructures demonstrates that the VLS growth mechanism is maintained even under the nitrogen flux. Preservation of the VLS-growth mechanism provides a relatively high growth rate (3 μm/h) of N-containing NW segments. It is found that the incorporated nitrogen has a minor impact on the NW crystal structure, which cannot be resolved by the TEM imaging as well as on the morphology. However, the formation of axial heterostructure was resolved in micro-Raman spectroscopy by the observation of the spatially localized Raman-active alloying mode.

Embedding of the nanowire array into the silicone rubber membrane and further release from the growth substrate allows to study the optical properties of the NW array apart from and parasitically grown GaPN layer and Si. Study of the PL response from the as-grown structure, NW/PDMS membrane and the substrate after the NWs exfoliation demonstrates that nitrogen is incorporated both into NWs during their VLS growth and into parasitically grown nanoislands with different content. It was found that nitrogen incorporates 3–5 times more efficiently into NWs (concentration of 0.9–1%) in comparison with the island layer (concentration of 0.2–0.3%). The effective incorporation of nitrogen during the VLS growth of NWs makes it possible to create the axial NW heterostructures, which is of great practical importance.

## Figures and Tables

**Figure 1 nanomaterials-10-02110-f001:**
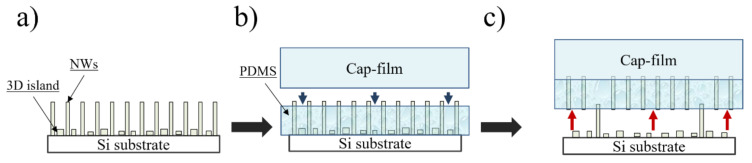
Schematic representation of the (**a**) nanowire (NW) as-grown array on Si substrate, (**b**) encapsulation of NW array into polydimethylsiloxane (PDMS) membrane and cap-film covering, (**c**) the NW/PDMS membrane release.

**Figure 2 nanomaterials-10-02110-f002:**
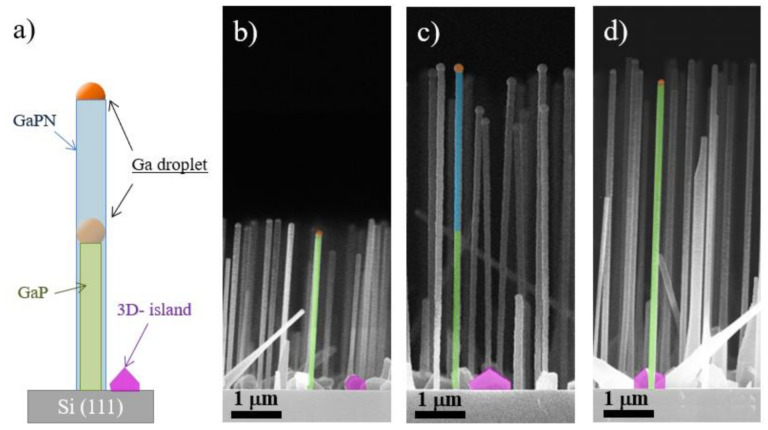
(**a**) Schematic representation of the NW geometries. Cross-sectional view SEM images of: (**b**) GaP stems, (**c**) heterostructured GaPN/GaP NWs, and (**d**) GaP NW array (reference sample).

**Figure 3 nanomaterials-10-02110-f003:**
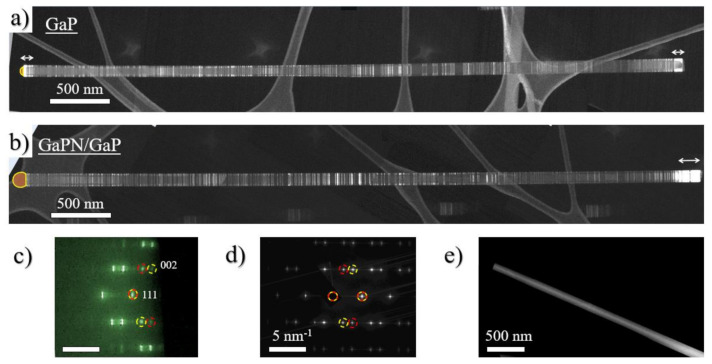
Dark-field TEM images of the studied (**a**) GaP reference NW, (**b**) heterostructured GaPN/GaP NW wurtzite (WZ) insertions are depicted by white arrows, (**c**) reflection high energy electron diffraction (RHEED) pattern (**d**) selective area electron microdiffraction pattern and (**e**) HAADF scanning transmission electron microscopy (STEM) image of heterostructured GaPN/GaP NW.

**Figure 4 nanomaterials-10-02110-f004:**
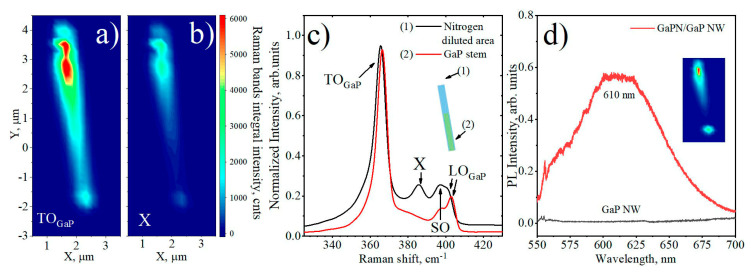
Raman spectroscopy mapping of individual exfoliated heterostructured GaPN/GaP NW transferred onto the glass substrate: spatial distribution of integrated scattering intensity of the (**a**) transverse optical (TO)-GaP and (**b**) GaPN-specific alloying X bands; (**c**) typical Raman spectra obtained at the top and the bottom part of NW (NW position schematically presented in the inset); (**d**) PL spectra taken from heterostructured GaPN/GaP and pure GaP NW, the inset is an integral PL intensity spatial map taken for the same GaPN/GaP NW.

**Figure 5 nanomaterials-10-02110-f005:**
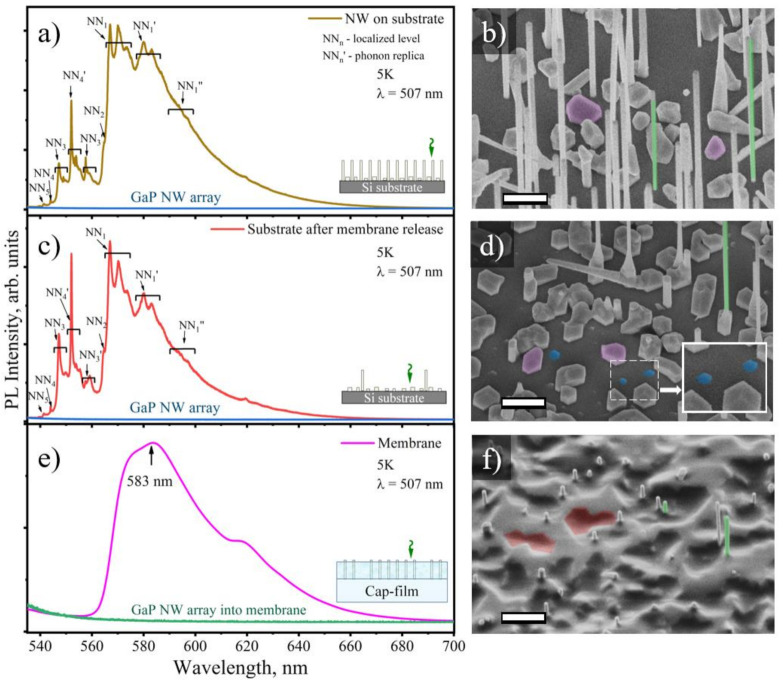
(**a**,**c**,**e**) Low-temperature PL response measured on (**a**) the as-grown GaPN/GaP NWs array on Si(111), (**c**) the growth substrate after membrane release, (**e**) the exfoliated NW/membrane (insets illustrate experiment geometry); (**b**,**d**,**f**) corresponding inclined-view (45°) sample surface SEM images (scale bar—1 µm), corresponding zoomed area presented in inset of (**d**).

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
