# Peer review of "Structural and Optical Properties of Self-Catalyzed Axially Heterostructured GaPN/GaP Nanowires Embedded into a Flexible Silicone Membrane"

_nanomaterials, 2020, doi:10.3390/nano10112110_

Round 1
Reviewer 1 Report
This manuscript presents a study on the self-catalyzed growth of axially heterostructured GaPN/GaP nanowires. The crystal structure and growth mechanism were discussed, and the structural and optical properties (Raman and PL) are also studied in order to study the growth mechanism. It is challenging to incorporate N into the nanowires, so this work represents a good progress. The results and discussion are interesting and sound, leading to solid conclusion. I would like to recommend it for publication with some minor revisions.
Comments:
1. The authors used the shift of Raman modes to study the spatial distribution of the NW chemical composition. Why not use high resolution EDS mapping? High resolution EDS mapping was used to observe the chemical composition of WO3 NWs, pls refer to paper - Nanotechnology, 31, 274003 (2020). High resolution EDS mapping should be able to provide direct information on the spatial distribution of the NW chemical composition.
2. When discussing the self-catalyzed growth (section 3.1), Ga droplets are obviously used as the seeds for the growth. Similar self-catalyst growth was observed before when growing InSb NWs on InAs, pls refer to the paper - Nanoscale Research Letters, 8, 333 (2013). This can be an evidence.
3. Can the authors provide more information on the NW growth? Obviously, growth parameters such as growth temperature will have influence on the NW morphology such as NW density, and length, which will directly affect their final device applications. Such influence was observed for InAs NWs before, pls refer to paper - Nanoscale Research Letters, 6, 463 (2011). Can the authors provide more information on this?
4. There are a quite number of typos in this manuscript. Should use superscript/subscript in many places, especially sections 2.1 and 2.2.
5. Figures: the font size of a), b), ... in some figures is a bit too big in comparison to that of other text in the figures.
6. The title of sections 2.1, 2.2, and 2.3 are the same. This might be typos. The authors should correct them.
Reviewer 2 Report
First of all, the authors claimed that the proposed approach of the heterostructured NW arrays synthesis and further free-standing NW/PDMS membrane fabrication can be used to create flexible orange-yellow light-emitting devices. However, the flexible OLED displays or recent quantum dots displays have more mature than the proposed NW fabricated by MBE. It should be deleted in this manuscript.
Secondly, despite a large number of works dedicated to growth and study of diluted nitride NWs, several fundamental questions are still unraveled. With the main being, whether it is possible to obtain VLS growth of the GaPN. This article showed the PA-MBE growth of axially heterostructured diluted nitride GaP/GaPN NWs and investigated their corresponding structural and optical properties. On of the weak point of this manuscript is the lack of NW density in cm-2. It would be more reasonable if the manipulation of NW density can be addressed in the revised manuscript.
Round 2
Reviewer 2 Report
The manuscript has been revised based on the reviewer's comments and shows more clear content as well as the density of nano wires. It can now be accepted from my point of view.